# A sex-stratified analysis of the genetic architecture of human brain anatomy

Rebecca Shafee [1] ✉, Dustin Moraczewski [2], Siyuan Liu [1], Travis Mallard [3,4], Adam Thomas [2] & Armin Raznahan [1] ✉

Large biobanks have dramatically advanced our understanding of genetic influences on human brain anatomy. However, most studies have combined rather than compared male and female participants. Here we screen for sex differences in the common genetic architecture of over 1000 neuroanatomical phenotypes in the UK Biobank and establish a general concordance between male and female participants in heritability estimates, genetic correlations, and variant-level effects. Notable exceptions include higher mean heritability in the female group for regional volume and surface area phenotypes; between-sex genetic correlations that are significantly below 1 in the insula and parietal cortex; and a common variant with stronger effect in male participants mapping to *RBFOX1* - a gene linked to multiple neuropsychiatric disorders more common in men. This work suggests that common variant influences on human brain anatomy are largely consistent between males and females, with a few exceptions that will guide future research in growing datasets.

Our understanding of genetic influences on human brain anatomy has expanded rapidly in recent years[1–3] due to the availability of combined neuroimaging and genetic information in large datasets such as the UK Biobank [UKB[4]] and international consortia [e.g., ENIGMA[5]]. Collectively, the rapidly growing number of studies in such datasets has: established the high heritability of many neuroanatomical phenotypes; revealed regional variation in the heritability and genetic architecture across different features of the brain; identified sets of genetic variants that shape different global and regional aspects of brain anatomy; and, established overlaps between the genetic determinants of brain anatomy and risk for brain-based neuropsychiatric disorders[1,2,6]. However, with few exceptions[2,7], this growing and impactful literature has typically combined the male and female participants rather than directly comparing genetic influences on brain anatomy between the two groups.

Several observations strongly motivate comparing the genetic architecture of neuroanatomical variation in males and females. Structural magnetic resonance imaging (sMRI) studies of brain anatomy have identified several reproducible sex differences in brain anatomy, including greater mean total brain volume in male compared to female participants[8,9], which survives statistical correction for sex differences in height[10], and sex differences in regional brain anatomy above and beyond these differences in overall brain size[9,11–13]. If these phenotypic sex differences partly reflect sex-specific biological influences on brain development, then this would provide an opportunity for sex-differentiated genetic influences on brain anatomy. In support of this reasoning, a large corpus of experimental research in animal models indicates that several canonical sex differences in regional anatomy of the mammalian brain can indeed be determined by sex-specific influences of gonadal steroids and sex chromosomes[12,14–20]. For example, because several regions of sex-differentiated brain volume in rodents are established by the effect of testosterone (via aromatization to estradiol) on apoptosis in male rodents only[21] - genetic variation in the strength of these influences would be predicted to modulate interindividual variation of region size in male rodents more prominently than females. In a similar fashion, all placental mammals show sex differences in the dosage of X and Y chromosomes (males XY and females XX) - which contain genes that are known to influence regional brain anatomy[22–24] and thereby introduce sex-specific genetic sources of neuroanatomical variation. Finally, the potential for sex differences

[1]Section on Developmental Neurogenomics, Human Genetics Branch, NIMH Intramural Research Program, NIH, Bethesda, MD, USA. [2]Data Science and Sharing Team, NIMH, NIH, Bethesda, MD, USA. [3]Psychiatric and Neurodevelopmental Genetics Unit, Center for Genomic Medicine, Massachusetts General Hospital, Boston, MA, USA. [4]Department of Psychiatry, Harvard Medical School, Boston, USA. ✉e-mail: rebecca.shafee@nih.gov; raznahana@mail.nih.gov

in genetic influences on neuroanatomical variation is also suggested by the observation that neuroanatomical correlates of several heritable neuropsychiatric disorders have been reported to differ between males and females[25,26], which could arise if disease-relevant genetic variants differentially influenced brain anatomy as a function of sex. Despite these numerous grounds for hypothesizing sex differences in the architecture of genetic influences on brain anatomy - we have so far lacked any direct tests for such differences in humans.

Here, we use a sample from the UK Biobank (UKB, 14534 male and 16294 female participants, Supplementary Data 1 and Fig. 1) to systematically compare the genetic architecture of neuroanatomical variation in biologically male and female individuals (participants with XY and XX karyotypes defined as male and female, respectively). We examine 1106 phenotypes including: 1080 regional measures of cortical anatomy encompassing estimates of gray matter volume (GMV), surface area (SA), and cortical thickness (CT) for 360 regions of interest[27] (each corrected for the corresponding global brain phenotype); 23 subcortical structure volumes (Supplementary Data 1, corrected for total brain volume), and 3 global measures (mean cortical thickness, total cortical surface area, and total brain volume). We distinguish between these different morphometric properties of the brain because they are known to show distinct genetic architectures[1,2,6] and varying mean differences between males and females[9,11]. We first compare the total SNP-heritability ($h^2_{SNP}$, autosomal and X-chromosomal; SNP = Single Nucleotide Polymorphism) of each phenotype between the male and female participants. It should be noted that estimated SNP heritability is usually less than heritability estimates from twin studies. Next, we evaluate genetic correlations ($r_g$) between the male and female participants for each phenotype, screening for any potential instances where this correlation differs from 1. Finally, we carry out sex-stratified genome-wide association analyses (GWAS) for each phenotype to test for any genetic variants with significantly different between-group effects.

Our systematic screen finds that the genetic architecture of neuroanatomical variation in humans is broadly congruent between males and females, but also highlights three notable sex differences. First, there is a general tendency across all brain regions for female participants to show higher mean SNP-based heritability (mean $h^2_{SNP}$) for cortical GMV and SA than males. Second, we find that the strength of genetic correlations ($r_g$) between male and female participants varies substantially across the cortical sheet and falls significantly below 1 for isolated regions of the parietal cortex and insula. Finally, after stringent control for multiple comparisons across all brain regions, we find

statistically significant sex differences in the phenotypic effects of a common variant mapping to *RBFOX1* - a known risk gene for neuropsychiatric and neurodevelopmental disorders like Autism Spectrum Disorder (ASD) and schizophrenia which are more common in one sex compared to the other[28,29]. Across all phenotypes, 571 more genes show evidence for sex-differentiated SNP effects at a genome-wide level of statistical significance ($p < 5e-8$), and some of these nominal associations may reach statistical significance with expanding sample sizes[30]. Taken together, these results provide a benchmark view of sex differences in the genetic architecture of human brain anatomy, which points toward general convergence between male and female participant groups but also highlights important instances of divergence that warrant repeated investigation as biobank sample sizes increase.

## Results
### Sex-difference in SNP-based heritability
Most brain phenotypes are significantly heritable, and it is possible to estimate the fraction of the phenotypic variance that is captured by genotyped SNPs using the genomic relationship matrix (GRM) approach. Using GCTA[31], we constructed sex-specific autosomal and X-chromosomal GRMs from which SNP-heritability ($h^2_{SNP}$) was estimated in each sex for: 180 regional measures (HCP parcellation[27]) of GMV, SA, and CT in each hemisphere; 23 subcortical volumes; and, 3 global measures (total 1106 phenotypes).

Figure 2 shows the spatial maps of total $h^2_{SNP}$ for cortical GMV, SA, and CT in each sex (Fig. 2a–c, respectively). Separate autosomal and X-chromosomal contributions to $h^2_{SNP}$ can be found in the Supplementary Data (2–4). X-linked heritability was estimated using the best-fit dosage compensation models (full dosage compensation/no dosage compensation/equal variance) from Mallard et al.[22] for each phenotype. The spatial variation in $h^2_{SNP}$ estimates was qualitatively similar between male and female participants for regional GMV, SA, and CT, and the between-sex correlation in $h^2_{SNP}$ across all brain regions was high for all three phenotypic classes ($r = 0.79$ for GMV, $r = 0.83$ for SA and $r = 0.67$ for CT; $p < 2.2e-16$ for all three). No statistically significant between-sex difference in $h^2_{SNP}$ was observed for any of these regional $h^2_{SNP}$ estimates ("Methods") after correcting for multiple-testing (MTC) ($p_{MTC} = 0.05/360 = 1.4e-4$; all $p > 1.4e-4$, Supplementary Data 2–4). In addition, cross-region variation in the magnitude of sex differences in $h^2_{SNP}$ (Fig. 2g–i) was unrelated to the magnitude of phenotypic sex differences (calculated as described in Methods) for GMV and SA but showed a weakly negative correlation

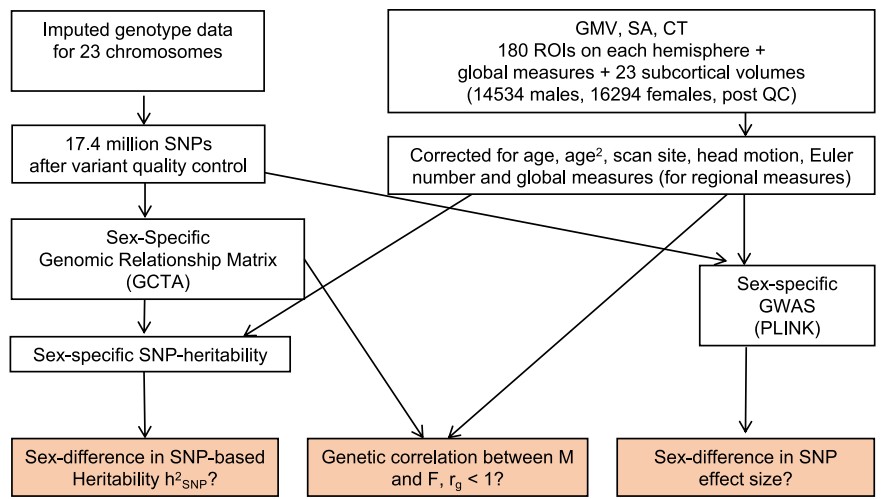

**Fig. 1 | Analysis of sex-difference in the genetic architecture of brain MRI phenotypes in the UK Biobank.** Outline of steps followed to answer the three main questions (shown in orange) addressed in this work.

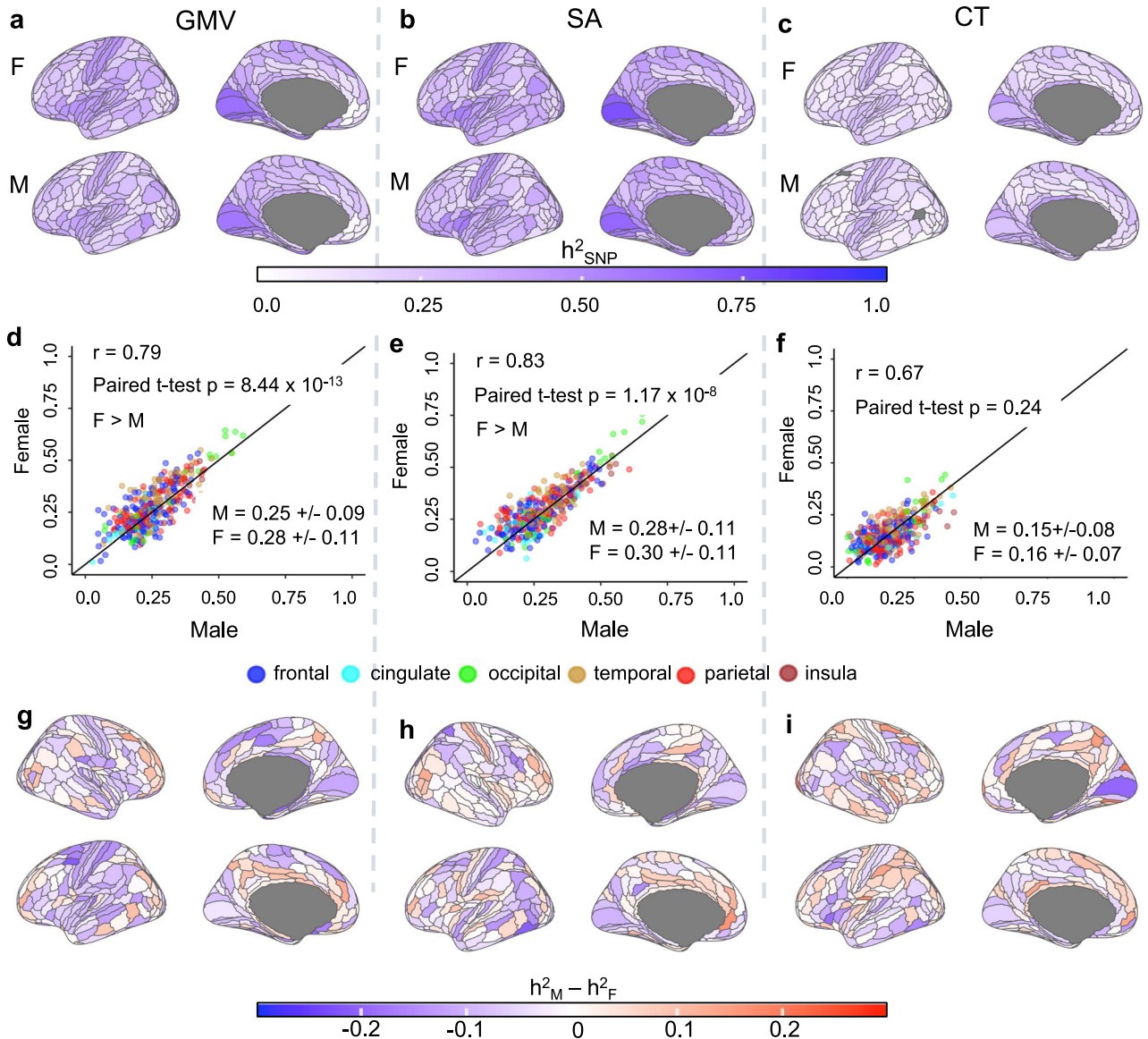

**Fig. 2 | Comparing SNP-based heritability, $h^2_{SNP}$, of regional gray matter volume (GMV), surface area (SA), and cortical thickness (CT) between male and female participants. a–c** Sex-specific (M: male, F: female) spatial maps of SNP-based heritability ($h^2_{SNP}$) of regional cortical GMV (**a**), SA (**b**) and CT (**c**). Only left hemispheres are shown - results for both hemispheres can be found in Supplementary Fig. 1. **d–f** Scatter plots of $h^2_{SNP}$ values of male and female participants across 360 cortical regions in the HCP parcellation ("Methods") for GMV (**d**), SA (**e**) and CT (**f**), respectively, with inset statistics – top: Pearson's correlation in $h^2_{SNP}$ across regions and p-values of two-sided paired *t* tests of regional $h^2_{SNP}$ between sexes (SA: *t* = − 5.83, df = 359, mean difference = − 0.019, 95% CI = [− 0.025, − 0.012]; GMV: *t* = − 7.42, df = 359, mean difference = − 0.02, 95% CI = [− 0.033, − 0.019]; CT: *t* = − 1.17, df = 359, mean difference = − 0.0039, 95% CI = [− 0.01, 0.002]); bottom: sex-specific mean and standard deviation. The colors of the circles correspond to the different lobes of the cortex as shown in the legends and the solid black line shows a line with a slope of 1. **g–i** Spatial map of sex-differences in $h^2_{SNP}$ ($h^2_{SNP\text{-male}}$ − $h^2_{SNP\text{-female}}$) for GMV, SA, and CT, respectively (all regions are shown; none show multiple-testing corrected (MTC) significant sex-difference corresponding to $p < 1.4e\text{-}4$, Supplementary Data 2–4). All phenotypes were corrected for corresponding global measures (mean thickness, total surface area, and total brain volume). Source data are provided as Source Data files.

with regional variation in phenotypic sex-differences for CT ($r = -0.14$, $p = 0.007$, Supplementary Fig. 2). However, when considering the distribution of $h^2_{SNP}$ across all cortical regions collectively using paired two-tailed *t* tests, mean $h^2_{SNP}$ was significantly higher in female participants than males for both GMV and SA ($t$ test $p = 8.4e\text{-}13$, $t = -7.42$, df = 359, mean difference = − 0.02, 95% CI = [− 0.033, − 0.019] for GMV and $p = 1.17e\text{-}8$, $t = -5.83$, df = 359, mean difference = − 0.019, 95% CI = [− 0.025, − 0.012] for SA; Wilcoxon rank test $p = 1.64e\text{-}11$ for GMV, $p = 1.4e\text{-}7$ for SA; Fig. 2d–f). We observed moderate to high total $h^2_{SNP}$ for all 23 subcortical volumes and all three global measures, which did not differ significantly between the sexes (Supplementary Data 5, 6 and Supplementary Fig. 3).

Since, by definition, $h^2_{SNP}$ is the ratio of genetic variance $V_G$ and phenotypic variance $V_P$, where $V_P = V_G + V_E$, ($V_E$ is the residual variance not attributable to additive genetic effects), we sought to refine the above finding - of greater mean $h^2_{SNP}$ in female vs. male participants for regional GMV and SA measures - by examining $h^2_{SNP}$, $V_G$, $V_P$ and $V_E$ in both sexes. Paired two-tailed *t* tests (as well as nonparametric Wilcoxon rank tests) between the sexes indicated higher mean $V_G$, $V_P$, and $V_E$ in male participants compared to female participants in GMV ($V_G$: $t = 5.99$, df = 359, $p = 5.02e\text{-}9$; $V_P$: $t = 12.38$, df = 359, $p < 2.2e\text{-}16$; $V_E$: $t = 12.30$, df = 359, $p < 2.2e\text{-}16$) and SA ($V_G$: $t = 7.13$, df = 359, $p = 5.2e\text{-}12$; $V_P$: $t = 9.74$, df = 359, $p < 2.2e\text{-}16$; $V_E$: $t = 9.44$, df = 359, $p < 2.2e\text{-}16$), but not CT ($V_G$: $t = -1.92$, df = 359, $p = 0.055$; $V_P$: $t = -0.63$, df = 359,

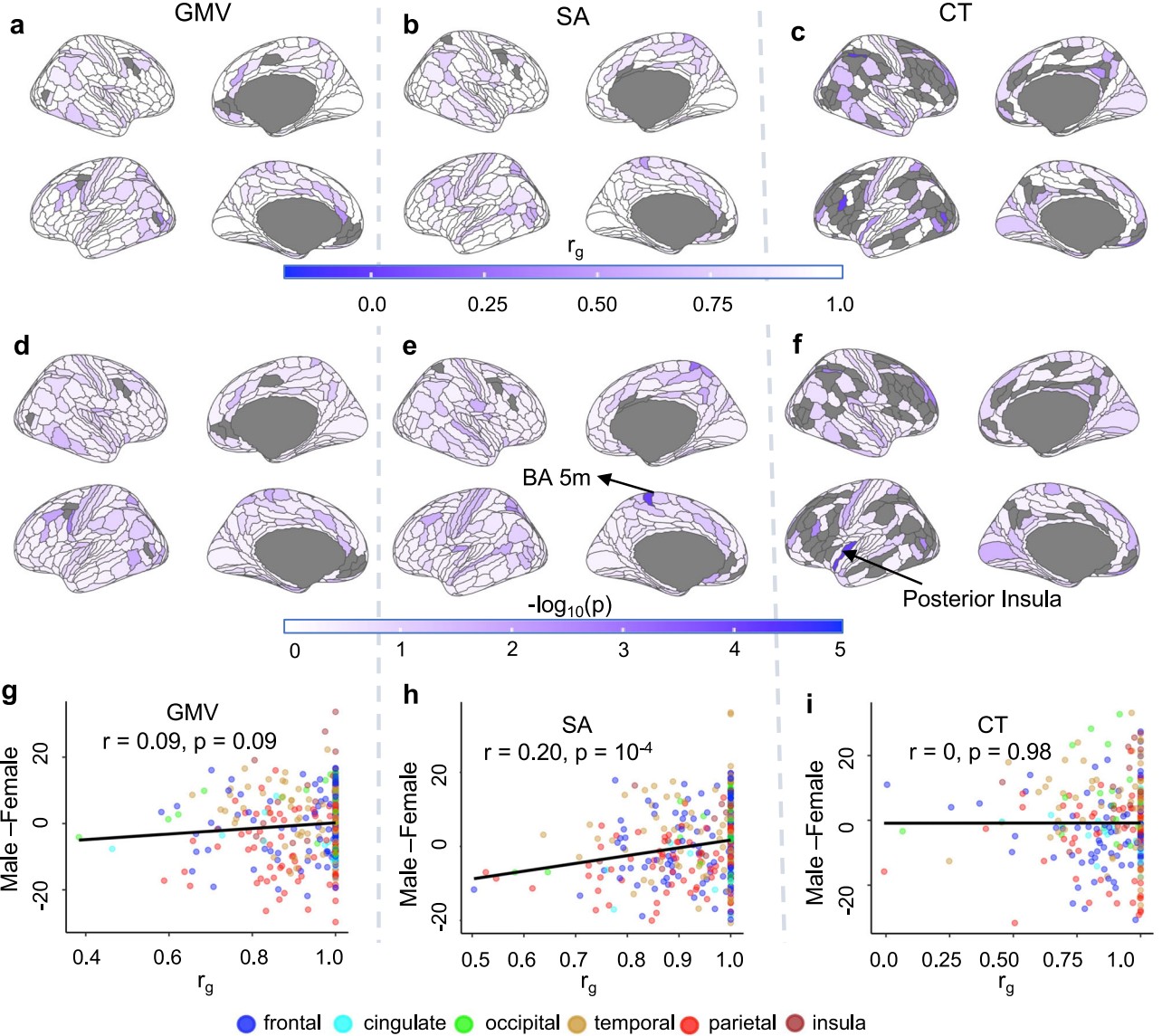

**Fig. 3 | Between-sex genetic correlations ($r_g$) in brain anatomy. a–c** Genetic correlation ($r_g$) across all autosomes between male and female participants in regional gray matter volume, GMV (**a**), surface area, SA (**b**) and cortical thickness, CT (**c**) calculated using GCTA. Regions with low heritability ($p > 0.05$ for $h^2_{SNP}$ in either sex) were excluded in this step and are shown in gray. **d–f** $p$-values (-log10(p)) of log-likelihood tests of $r_g < 1$ in GCTA for regional GMV (**d**), SA (**e**) and CT (**f**). Exact $p$-values are reported in the Source Data file. Only two regions showed significant $r_g < 1$ after multiple-testing correction (MTC) ($p_{MTC} = 1.4e\text{-}4$): one in SA (superior parietal lobule medial Brodmann area 5, BA 5m, $p = 7.9 \times 10^{-5}$, $r_g = 0.50 +/- 0.11$) and one in CT (posterior insula, PoI1, $p = 7.7e\text{-}5$, $r_g = 0.46 +/- 0.11$). All regional

phenotypes were corrected for corresponding global measures (mean thickness, total surface area, and total brain volume) prior to calculating $r_g$. **g–i** Comparison of phenotypic sex difference and genetic correlation, $r_g$, in GMV (**g**), SA (**h**), and CT (**i**) for each region of the HCP parcellation with non-negligible $h^2_{SNP}$ in both sexes (i.e., with $p < 0.05$ for $h^2_{SNP}$). Pearson's correlation coefficients and corresponding $p$-values are reported for each category of phenotype. The colors of the circles represent the different lobes of the cortex, as shown in the legend. Phenotypic sex differences are shown in $t$-statistics ("Methods"), with positive values indicating higher in male participants. Results for gray regions (**a–f**) can be found in Supplementary Fig. 5. Source data are provided as a Source Data file.

$p = 0.52$; $V_E$: $t = 0.39$, df $= 359$, $p = 0.69$; Supplementary Data 7). Thus, consistent with previously reported results[8], male participants showed higher phenotypic and genetic variance in GMV and SA averaged across all brain regions even though, on average, heritability, which is the ratio of the two, was higher in female participants (Supplementary Fig. 4 and Supplementary Data 8).

**Between-sex genetic correlation in brain anatomy**

For each phenotype, we used the "bivariate" option in GCTA to calculate the genetic correlation ($r_g$) between male and female participants. Since low $h^2_{SNP}$ can make the estimation of $r_g$ unstable, we limited $r_g$ calculations to phenotypes with $h^2_{SNP}$ $p$-values of 0.05 or

lower in both sexes (all subcortical and global phenotypes; 346 GMV regional phenotypes, 352 regional SA phenotypes, and 255 regional CT phenotypes). This analysis was also restricted to autosomes (covering 95% of the genome) because of the relatively small X-chromosome heritability contributions. Across cortical regions, the strength of between-sex $r_g$ varied substantially across regions (Fig. 3; GMV: 0.38 to 1; SA: 0.5 to 1; CT: 0.004 to 1), but median values were consistently high (GMV: 1; SA: 1; CT: 0.98). Likelihood tests to detect $r_g$ values less than 1 were not significant for most regions after multiple-testing correction (MTC) (Fig. 3 and Supplementary Data 9–11) - indicating broad between-sex similarity in the common genetic architecture of cortical anatomy. Only two regional phenotypes possessed between-sex

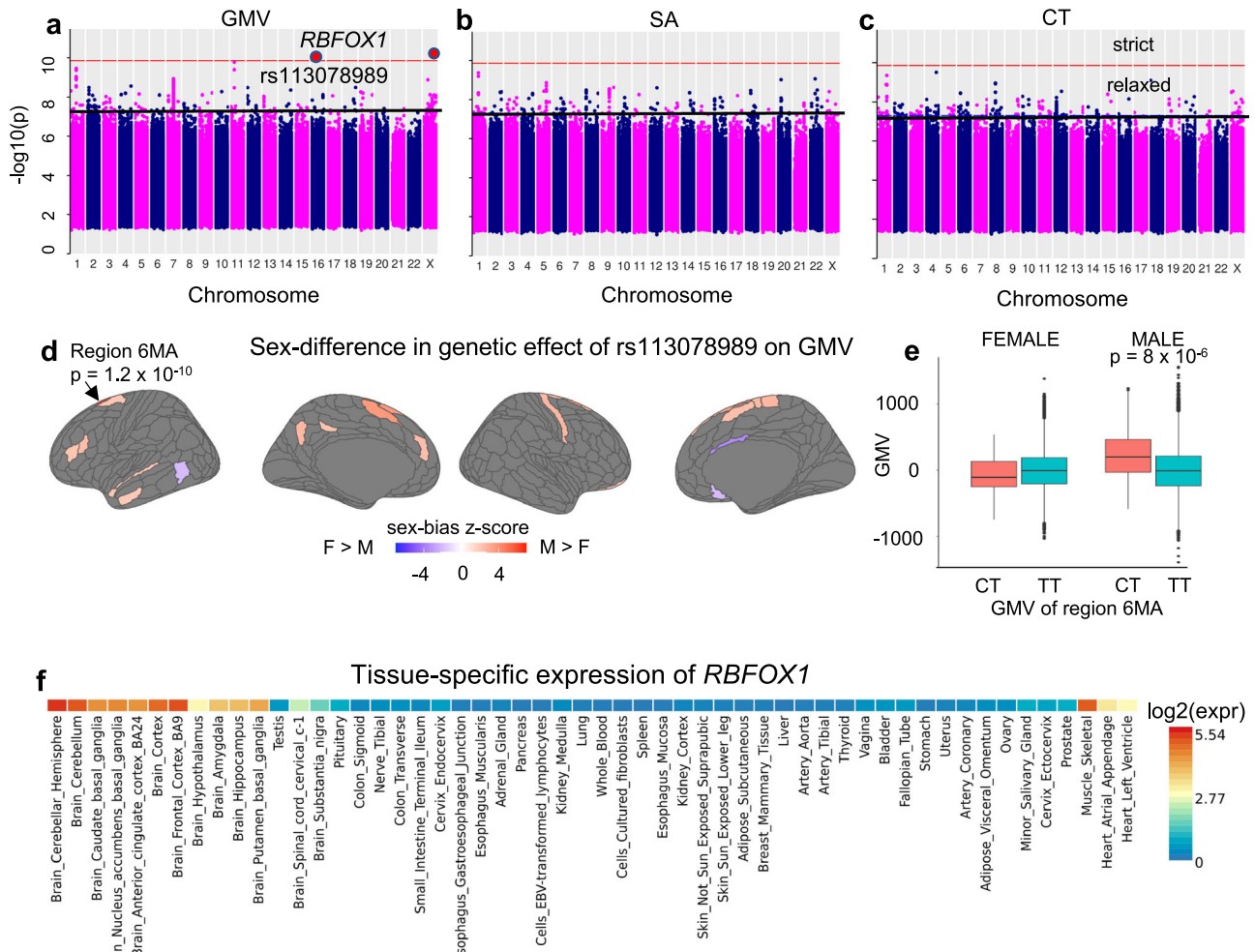

**Fig. 4 | Genome-wide tests of sex-difference in SNP effects in regional gray matter volume (GMV), surface area (SA), and cortical thickness (CT).**
**a**–**c** Minimum *p*-value across all 360 cortical regions for each SNP for each phenotype category: GMV (**a**), SA (**b**), and CT (**c**). The red and the black horizontal lines correspond to the "strict" and the "relaxed" significance thresholds ($p_{strict} = 1.4e-10$, which accounts for multiple-testing and $p_{relaxed} = 5e-8$, respectively). Red circles indicate SNPs above the "strict" threshold: chr16:rs113078989 ($p = 1.2e-10$) and chrX:rs747862348 ($p = 9.5e-11$). rs113078989 mapped to protein-coding gene *RBFOX1*. The sample sizes for the relevant sex-stratified genome-wide association analyses may be found in Supplementary Data 1. **d** Sex-difference in the effect of rs113078989 across the cortex on cortical GMV shown as z-scores ("Methods") with red indicating higher magnitude in male participants; all regions with sex-difference *p*-value > 0.05 are shown in gray. **e** Boxplot showing GMV values for each

sex ($N = 13968$ male, $N = 15613$ female) for the CT and TT genotypes of rs113078989 in region 6MA. The *p*-value for the difference in GMV between the two genotypes in the male group was calculated using a two-sided *t* test ($N = 13968$, $t = -4.7$, df = 81.7, $p = 8.2e-6$, mean difference = $-214.17$, 95% CI = [$-303.67$, $-124.68$]). In the boxplots, the center shows the median, and the lower and the upper hinges correspond to the first and third quartiles. The lower whisker extends from the hinge to the smallest value at most 1.5 * IQR of the hinge, where IQR is the distance between the first and the third quartile. The upper whisker extends from the hinge to the largest value no further than 1.5 * IQR from the hinge. **f** GTEx tissue expression heatmap plot for *RBFOX1* indicating significant brain expression (from FUMA). The unit shown is log2 of the expression per label. Red indicates high expression. Source data are provided as Source Data files.

$r_g$ values that were statistically significantly lower than one ($p < p_{MTC}$, Fig. 3): CT of left inferior posterior insula (PoI1, $p = 7.7e-5$, $r_g = 0.46 +/- 0.11$) and left superior parietal lobule medial Brodmann Area 5 SA (BA 5m, $p = 7.9 \times 10^{-5}$, $r_g = 0.50 +/- 0.11$). Of note, these two phenotypes also showed statistically significant sex differences in mean values within the UKB - SA of medial BA 5 region was higher in females, and CT of PoI1 was higher in males (Supplementary Fig. 2b, c). However, the general relationship between the magnitude of phenotypic sex differences (calculated as described in Methods, also Supplementary Fig. 2) and $r_g$ across the cortex was not significant for GMV and CT ($r = 0.09$ and $p = 0.09$ for GMV; $r = 0$, $p = 0.98$ for CT), but weakly positive for SA ($r = 0.20$, $p = 0.0001$), with lower $r_g$ regions having higher magnitude in female participants (Fig. 3).

Between-sex $r_g$ was statistically indistinguishable from 1 for all global and subcortical phenotypes (Supplementary Data 12, 13).

Increasing sample size in the future would help to estimate $r_g$ more accurately and, therefore, likely bring many of the regions with between-sex $r_g$ values substantially below 1 into statistical significance (e.g., GMV of left lateral occipital lobe, LO1: $r_g = 0.64 +/- 0.16$, SA of left middle temporal visual region, MST: $r_g = 0.62 +/- 0.18$, and CT of right middle temporal gyrus, TE1p: $r_g = 0.68 +/- 0.20$).

**Sex-difference in SNP effects**
We screened for individual SNPs with sex-differentiated effects on brain phenotypes using sex-stratified GWAS. For each SNP and each phenotype, the sex difference in effect sizes was calculated as a z-score through Eq. 1 ("Methods") using the sex-specific effect sizes and the corresponding standard errors from which a corresponding *p*-value was estimated assuming normal distribution in R ("Methods"). Significant sex differences were considered at two different thresholds: (i)

a "strict" threshold which accounted for multiple testing across brain regions (corrected $p$-value = standard genome-wide significance $p$-value/number of regions), and (ii) a "relaxed" threshold in keeping with prior work[32] corresponding to the standard genome-wide significance threshold of $p_{relaxed} < 5e\text{-}8$. For cortical phenotypes, all measures in each category (regional CT, SA, and GMV and corresponding global phenotypes) were grouped together for multiple testing corrections resulting in $p_{strict} < 5e\text{-}8/(361) = 1.4e\text{-}10$.

At the "strict" threshold, two SNPs showed statistically significant sex differences in their phenotypic effects on regional cortical GMV measures (chr16:rs113078989 in region Left 6MA, chrX:rs747862348 in region Left TE1p; Supplementary Data 14), and none for CT and SA phenotypes (Figs. 4a–c), or global and subcortical phenotypes. The SNPs underwent positional mapping (10 kb symmetric window) to protein-coding genes in MAGMA[33] as implemented in FUMA[34]. The autosomal SNP chr16:rs113078989 mapped to the intronic region of the gene *RBFOX1,* whereas chrX:rs747862348 was not within 100 kb of any gene and was not within any enhancer or promoter region (www.ucsc.edu, Supplementary Fig. 6).

Figure 4d shows cortical regions where rs113078989 showed sex-differentiated relationships with GMV at the $p < 0.05$ level of significance, including the supplementary motor region (6MA), where this association also reached the strict statistical threshold (i.e., correction across the genome and all cortical regions). For 23 of these 26 cortical regions showing suggestive sex-differentiated association between rs113078989 and GMV (including region 6MA) - the sex-difference reflected a more prominent association in male participants than female participants (shown as a boxplot for 6MA in Fig. 4e). This predominant stronger effect of rs113078989 on GMV in the male group was also apparent when considering all cortical regions collectively (mean sex-difference z-scores of rs113078989 was positive on both hemispheres: mean $z_{left} = 0.55$, mean $z_{right} = 0.15$; Wilcoxon rank test $p$-values for mean $z > 0$: $p_{left} = 7e\text{-}11$, $p_{right} = 0.04$). A test for potential enrichment of these regions using spin tests with the 17 networks of Yeo, Krienen et al.[35] pointed to suggestive overlap ($p = 0.07$) with the default mode network (DMN-2) for the M > F regions and no overlap with the functional networks for the F > M regions (Supplementary Data 21). Tissue-specific expression (Fig. 4f) from GTEx data (V8, as implemented in FUMA) indicated high expression in the brain for *RBFOX1* with additional muscular expression.

SNPs passing the "relaxed" threshold mapped to 572 unique genes (108 SNPs mapped to 226 genes in GMV, 108 SNPs mapped to 177 genes in SA, 123 SNPS mapped to 173 genes in CT, 4 SNPS in subcortical volumes mapped to 13 genes, Supplementary Data 15–19). No SNP passed the "relaxed" threshold for the three global phenotypes. Functional analyses with GENE2FUNC in FUMA did not identify any statistically significant molecular function, biological process, or cellular compartment GO term enrichments for genes identified using the relaxed threshold - regardless of whether mapped genes for each phenotype were considered as separate sets (i.e., regional cortical GMV, SA, CT, and subcortical volumes), or combined into a unique gene set (comprising 338 independently significant SNPs which mapped to 572 genes, Supplementary Data 20), or whether the background gene set used for enrichment analysis was the default all gene option in FUMA or a set of brain-specific genes[36].

## Discussion

Our study - which represents a systematic survey for potential sex differences in the common variant genetic architecture of human neuroanatomical variation - generated several findings of note, which are considered in turn below, along with important caveats and limitations.

First, we found that several previously reported observations regarding SNP-based $h^2_{SNP}$ of brain anatomy when combining male and female participants[1–3,6,22,37] could be independently replicated in both sexes. Within both male and female groups separately, we see that heritability estimates for mean cortical thickness were lower than heritability estimates for total surface area or total GMV[2]; Supplementary Data 5). We also found that in each sex, some of the highest $h^2_{SNP}$ values for regional cortical measures are seen in the primary and secondary visual cortex regions for GMV and SA. In CT, we found the highest $h^2_{SNP}$ in the retrosplenial cortex in male participants and in V2 in female participants. The high heritability of the visual cortex has been previously reported in both twin studies (Strike et al.[38] reported the highest genetic contribution to phenotypic variance in visual cortex SA) and population-based studies[1,2,6]. The lowest values of $h^2_{SNP}$ of GMV and SA were seen in areas including the rostral anterior cingulate cortex, medial prefrontal area, and frontal eye field - regions which are also reported to have some of the lowest $h^2_{SNP}$ values in Smith et al.[2]. Similarly, of the 23 subcortical volumes examined, we found within both sexes that $h^2_{SNP}$ values were highest for the cerebellum, putamen, and caudate nucleus and lowest for the amygdala (Supplementary Data 6) consistent with findings of Satizabal et al.[3] and Smith et al.[2]. In their recent work focusing on the effects of the X-chromosome on brain anatomy Jiang et al.[39] reported sex-stratified $h^2_{SNP}$ for CT, GMV and SA in the UK Biobank. Due to the difference in choice of parcellation we were not able to directly compare our results with their findings. However, their results also showed a high correlation between the $h^2_{SNP}$ of the male and the female groups for all three categories: $r = 0.82$ ($p = 3.8e\text{-}16$), 0.88 ($p < 2.2e\text{-}16$) and 0.71 ($p = 1.1e\text{-}10$) for GMV, SA and CT, respectively. Paired two-tailed $t$ tests using their sex-stratified $h^2_{SNP}$ values showed higher values in the male group in CT ($t = 4.0$, df = 61, $p = 0.00017$, mean difference = 0.027, 95% CI = [0.013, 0.041]) and in the female group in SA ($t = -3.56$, df = 61, $p = 0.00071$, mean difference = $-0.027$, 95% CI = [$-0.042$, $-0.012$]) - the latter in agreement with our findings although we did not detect any sex-difference in $h^2_{SNP}$ of CT.

Second, although direct group comparisons did not identify any individual anatomical phenotype with statistically significant sex differences in $h^2_{SNP}$ estimates, we did find statistically significant differences in the distribution of regional $h^2_{SNP}$ estimates for cortical GMV and SA, such that $h^2_{SNP}$ estimates were on average higher for female participants than male participants. This tendency towards greater trait $h^2_{SNP}$ in female participants has also been reported by some[32] prior sex-stratified analyses of heritability of non-neuroanatomical traits (e.g., for systolic, diastolic blood pressure and waist circumference[40,41]). Of the 31 traits considered in Gilks et al.[41], 15 showed higher $h^2_{SNP}$ in female participants, 3 in male participants, and 15 showed no sex difference. We showed that for the neuroanatomical phenotypes considered here, the observation of higher mean $h^2_{SNP}$ in female participants is accompanied by higher mean regional $V_G$ and $V_P$ values in male participants than females. Thus, for the neuroanatomical phenotypes examined here, there was a general tendency for greater phenotypic variance in male participants, whereas the proportion of phenotypic variance accounted for by the additive effect of common variants was generally greater in the female participants.

Third, since one of the motivations behind our work was the observed phenotypic sex-difference in brain phenotypes (e.g., Liu et al.[11]) we screened for significant association between phenotypic sex-difference and sex-difference in $h^2_{SNP}$ as well as between-sex $r_g$. We found no evidence of a significant correlation between phenotypic sex difference and sex difference in $h^2_{SNP}$ for GMV and SA, although a weak negative correlation was observed ($r = -0.14$, $p = 0.007$) for CT. In addition, there was no statistical evidence for a general trend whereby regions with greater sex differences in their mean values tend to show lower between sex genetic correlations in GMV or CT. However, a weak positive correlation was seen for SA ($r = 0.20$), with lower $r_g$ corresponding to regions with higher values in females. Notably, both of the

cortical phenotypes showing between-sex $r_g$ values significantly below 1 - BA5 surface area and insula cortical thickness - also showed significant phenotypic sex differences in their means. The lower between-sex $r_g$ in these regions suggests that differential genetic regulation of anatomical variation may contribute to sex differences in the mean trait value attained - although it remains unclear if these differences carry any functional consequences[12]. Nevertheless, other than these few exceptions, there was little evidence for a strong or general trend for regions of greater sex differences in phenotypic means showing larger sex differences in $h^2_{SNP}$ or lower between-sex $r_g$ values. Thus, in general, sex differences in the mean value of anatomical traits appear to be achieved by mechanisms that do not modify the collective magnitude or genome-wide distribution, of common variant influences on trait variation within each sex. Such mechanisms could potentially reflect sex differences in gene dosage or environmental exposures - which may both exert their effects in a developmentally dynamic manner.

Fourth, in keeping with high between-sex $r_g$ for almost all neuroanatomical phenotypes, we found very few SNPs (2 out of 12.7 million SNPS with MAF > 0.001) with sex-differentiated effects in GWAS after strict correction for multiple comparisons. With the aid of positional gene mapping (FUMA), we were able to map one of the SNPs (chr16:rs113078989) to *RBFOX1*- a synaptic gene expressed in both excitatory and inhibitory neurons (The human protein atlas, http://www.proteinatlas.org[42]). *RBFOX1* encodes a splicing factor important for neuronal development and has been previously implicated in several neurodevelopmental and neuropsychiatric disorders that are more common in men, including autism spectrum disorder (ASD), intellectual disability and epilepsy, attention-deficit hyperactivity disorder, schizoaffective disorder and schizophrenia[28,29]. It has also been found to show higher expression in females[43]. The *RBFOX1* implicating SNP rs113078989 has not been reported as a sex-differentiated eQTL[44], but we note that power is currently low for such analyses, even in GTEx. In our work, only the male group showed an effect of rs113078989 on the GMV of region 6MA, part of the supplementary motor region known to play a role in coordinating complex movements. Male carriers of the minor allele (CT genotype) showed higher GMV compared to homozygous individuals (TT genotype). The fact that the sole gene implicated by these analyses was strongly associated with such sex-differentiated conditions is certainly striking and points toward ways in which sex-specific genetic effects could potentially shape sex differences in the prevalence or presentation of neurodevelopmental disorders. However, it will be crucial to test the replicability of this finding as larger datasets for sex-stratified GWAS of neuroimaging traits become available. Work in these datasets may also bring some of the subthreshold SNPs from our relaxed threshold analyses into statistical significance - potentially expanding the number of genetic variants with sex-differentiated influences on human brain development. In addition, our finding of only two SNPs showing statistically significant sex-differentiated effects on brain anatomy is consistent with the recent suggestion by Zhu et al.[45] that gene-by-sex interactions may largely act through sex differences in the magnitude of many genetic effects ("amplification"), rather than differences in the identity of causal variants or the direction of their effects.

Our findings should be considered in light of several caveats and limitations - some of which may be addressable in future research as datasets continue to increase in size and diversity. First, the UKB dataset - although revolutionary in its impact - predominantly includes individuals of European descent between 40 and 80 years of age. As such, our findings cannot be assumed to generalize outside these demographic limits, and it will be crucial to revisit the questions addressed in our current study within different phases of the lifespan and in populations with more diverse genetic ancestries. Second, our study design does not include rare single nucleotide variants with MAF < 0.0003 (MAF < 0.001 for GWAS) or other classes of genetic variation such as indels or copy number variations - and future studies should also consider potentially sex-specific effects of these variant classes. Third, we have focused here on regional measures of brain anatomy using well-established parcellations of the human brain, but there are many alternative ways of measuring brain anatomy and many other imaging-derived phenotypes beyond those offered by structural MRI. Future studies should ideally extend to this broader range of phenotypes, although we note that the need for even more severe correction for multiple comparisons, and the lower measurement reproducibility for most imaging-derived phenotypes as compared to those structural MRI phenotypes we study here[46–48], will substantially lower statistical power unless sample sizes are dramatically increased beyond those included here. Fourth, while there are strong theoretical grounds (and some preliminary empirical findings herein) to motivate continued comparison of genetic influences on brain anatomy between males and females - future studies should also consider sex differences in environmental influences on the brain, and consider the many partly dissociable aspects of sex and gender that we are to some extent obscuring by the necessary treatment of sex as a binary variable in the present study.

Notwithstanding the above caveats and limitations, our study provides tests for potential sex differences in the genetic architecture of human brain anatomy using one of the largest available individual-level genotype and neuroimaging datasets. We investigate sex differences in brain-linked genetic measures at the individual SNP level as well as at the whole-genome level and find general concordance in the genetic basis of brain anatomical traits between male and female groups. Four notable exceptions to this general pattern are: (1) mean higher $h^2_{SNP}$ in the female group for GMV and SA but not CT; (2) two cortical regional phenotypes showing detectable deviation from $r_g = 1$; (3) weak spatial correlations between sex-differences in anatomy and sex differences in $h^2_{SNP}$ for CT, and lower $r_g$ values for SA; and, (4) preliminary evidence for a sex-specific relationship between neuroanatomy and common genetic variation mapping to *RBFOX1* - a gene implicated in the neurobiology of several psychiatric disorders more common in one sex compared to the other. The methods and results of this study - which represents a thorough systematic screen for sex differences in the genetic architecture of human neuroanatomical variation - offer a valuable reference point for future studies to be undertaken as available datasets increase in sample size, diversity of genetic ancestry, and phenotypic breadth.

## Methods

### Ethics
Our research complies with all relevant ethical regulations. The UK Biobank study was conducted under generic approval from the NHS National Research Ethics Service (ref 11/NW/0382). All participants provided informed consent. This study was conducted according to the guidelines in the Declaration of Helsinki, and all procedures involving human subjects/patients were approved by the North West Multi-center Research Ethics Committee.

### The UK biobank sample
The UKB is a large-scale biomedical study in which individuals were genotyped and completed questionnaires related to health, lifestyle, and environmental factors. A fraction of the participants also underwent brain MRI scans, which was the focus of our work. Further details of the UKB sample used in this work can be found in past studies[22] and also at https://www.ukbiobank.ac.uk. We used the most recent release of brain MRI data (downloaded on April 23, 2020) for 38,685 samples, together with the imputed genetic data provided by the UKB (Version 3) under application 22875. Our analyses included individuals with non-Hispanic European ancestry (according to UKB-provided information) to avoid population stratification-related confounding. In our study, sex was defined in terms of an individual's sex chromosomes: male if XY and female if XX. After imaging and genetic quality

control steps (as described below), the final data consisted of 14534 male participants and 16294 female participants (sample number varied with phenotype) and 17.38 million SNPs.

## Brain MRI phenotypes

Our study included regional cortical, regional subcortical, and global brain phenotypes. T1w images were processed with FreeSurfer v.6.0.0[49–51] to extract regional cortical gray matter volume (GMV), surface area (SA), and thickness (CT) using the multimodal HCP parcellation[27], which divides each hemisphere into 180 regions. Mean cortical thickness, total cortical surface area, and total brain volume were included as global measures. Lastly, 23 subcortical structure volumes (as calculated in the FreeSurfer pipeline) were included resulting in a total of 1106 phenotypes (Supplementary Data 1). Within each sex group, these phenotypes were corrected for age, age[2], head location in the scanner, scanning site, head motion (calculated from resting fMRI, provided by UKB), and Euler number[52], which is a measure of image quality. Regional cortical phenotypes were additionally corrected for respective global phenotypes: i.e., GMV for total brain volume, SA for total surface area, CT for mean cortical thickness, and subcortical structure volumes for total brain volume.

## Quality control

MRI image quality control steps included the removal of intracranial volume outliers (more than 4 SD away from the mean in each sex group) and samples with Euler number (reflecting image quality) < − 217[52], resulting in a total of 30827 individuals (14534 male participants and 16294 female participants).

Genetic data consisted of imputed genotypes available from the UKB (Version 3). Data for individuals passing MRI quality control steps as described above were extracted using PLINK v2.0[53]. Information provided by the UKB was used to remove individuals with putative sex chromosome aneuploidy, excessive heterozygosity, mismatched self-reported sex and genetic sex, and excessive relatedness. This was followed by the removal of variants with imputation INFO score < 0.3, MAF < 0.0003, Hardy-Weinberg equilibrium $p < 1e{-}6$, missingness > 0.05, and variants with more than two alleles. For the X-chromosome non-pseudoautosomal (non-PAR) region, these filters were applied to male and female participants separately and variants passing quality control in both sexes were retained for analyses. Lastly, individuals with missingness > 0.1 and one person from each pair of individuals with relatedness > 0.05 (as described in the next section) were also removed. The final set of genetic variants included 16.79 million autosomal and 585,465 X-chromosome SNPs. For sex-stratified GWAS and sex-differentiated SNP effect calculation, we used a more stringent MAF > 0.001 threshold[2,6] resulting in a total of 12.1 million autosomal SNPs and 584816 X-chromosome SNPs. For the X-chromosome, the MAF of SNPs was calculated separately for male and female participants and only SNP with MAF > 0.001 in both sexes were retained.

For each category (GMV, SA, and CT) of regional cortical phenotype, only individuals with all phenotypes in each hemisphere within 5 SD of the mean were included in the genetic analyses resulting in a small variation in the final sample size for genetic analysis in each category as shown in Supplementary Data 1 (for example, for GMV $N = 14087$ male participants and $N = 15771$ female participants).

## SNP-heritability, genetic correlation, and genome-wide association

Sex-specific SNP-based heritability ($h^2_{SNP}$) of each phenotype was estimated using sex-specific genomic relatedness matrices (GRMs) in GCTA[31] reflecting genetic similarity between pairs of individuals. GRMs were constructed for the autosomes and the X-chromosome separately in GCTA v1.93[31]. To ensure closely related individuals were not included in the analyses, the autosomal GRM was used to exclude one individual from each pair with relatedness > 0.05. The total SNP-heritability for each phenotype in each sex was estimated using a joint model including both the autosomal and X-chromosome GRMs. P-values corresponding to the significance of $h^2_{SNP}$ were estimated using likelihood ratio tests implemented in GCTA. The first 10 genetic ancestry principal components were included as covariates for the heritability analyses. For each phenotype, X-linked $h^2_{SNP}$ was estimated using the best-fit dosage compensation model from Mallard et al. [22].

For a trait, genetic correlation $r_g$ between the sexes was estimated also using GCTA. Since $r_g$ estimation can be unstable for low trait heritability, only traits with $h^2_{SNP}$ p-value < 0.05 were used. For this, a modified GRM approach was used following Yang et al.[54]: autosomal GRMs were calculated by combining the male and female participants, and modified phenotype files with two columns corresponding to the male and the female phenotypes were used with the "--reml-bivar" option. Deviation of $r_g$ from 1 was estimated using the "--reml-bivar-rg 1" option. We did not estimate $r_g$ for the X-chromosome due to a lack of statistical power (the highest $h^2_{X\text{-}chr}$ in GMV was 0.034 +/− 0.02 in male participants). For $r_g$ calculation also the first 10 ancestry PCs were used as covariates. Sex-stratified genome-wide association analyses (GWA - encompassing autosomes and the X-chromosome non-PAR region) for each phenotype were performed in PLINK V2.0[53] using the --linear option and the 10 ancestry PCs as covariates. Cortical surface plots were created using the "ggseg" package[55] in ref. 56.

## Estimating sex-differences

For each phenotype sex difference in total $h^2_{SNP}$ was estimated by calculating the following z-score:

$$Z = \frac{(X_M^2 - X_F^2)}{\sqrt{(SE_M^2) + (SE_F^2)}} \tag{1}$$

where $X_M^2$ and $X_F^2$ are $h^2_{SNP}$ of male and female participants, and $SE_M$ and $SE_F$ are the standard errors of the respective $h^2_{SNP}$ estimates (similar to the approach of Martin et al. [57]). Corresponding p-values were then calculated as $p = 2 \times (1 - \Phi(|Z|))$, where $\Phi$ is the cumulative distribution function of the standard normal distribution. For the regional cortical phenotypes, we report any result as significant for $p < p_{MTC} = 1.4e{-}4$, correcting for the number of cortical regions. For subcortical structures, we used a significance threshold of $p < 0.0021$ to correct for the 23 volumes tested.

Sex differences in $V_G$, $V_P$, and $V_E$ (additive genetic, phenotypic, and environmental variance, respectively) were tested using paired two-tailed $t$ tests as well as the Wilcoxon rank test in R. Phenotypic sex-difference was estimated by testing the significance of the coefficient "b" of "sex" in the linear model ("lm" function in R):

$$pheno = a + b * sex + c * age + d * age^2 + other\ covariates \tag{2}$$

where "pheno" corresponds to a GMV, SA, or CT phenotype, and "other covariates" are described earlier in Methods. The potential sex difference in the relationship between $V_P$ and $V_G$ was explored (Supplementary Fig. 4, Supplementary Data 8) by fitting the data to two models (moving to the simpler Eq. (4) in the absence of evidence for significant quadratic effects from Eq. (3)) shown below and testing the significance of the coefficients "d" and "f":

$$Model\ 1 : V_G = a + b * V_P + c * sex + d * sex * V_p + e * V_P^2 + f * sex * V_P^2 \tag{3}$$

$$Model\ 2 : V_G = a + b * V_P + c * sex + d * sex * V_p \tag{4}$$

Sex-difference in SNP effect size was estimated in a similar way for each variant and each phenotype using Eq. (1) where $X_M^2$ and $X_F^2$

represented GWAS effect sizes in male and female participants for that phenotype and $SE_M$ and $SE_F$ represented the corresponding standard errors of the effect sizes. The $p$-values calculated from these z-scores represented the significance of sex differences in SNP effects. We set two thresholds for downstream analyses: (i) a "strict" threshold correcting for multiple phenotypes by setting the significance threshold to $p_{STRICT} < 1.4e-10$, and (ii) a "relaxed" threshold using the standard genome-wide significance level of $p_{RELAXED} < 5e-8$. For subcortical structures, we set the "strict" threshold to be $p < 2.17e-9$, corresponding to correction for 23 phenotypes.

## Gene mapping and functional analyses

SNPs with significant (strict or relaxed) sex differences in effect sizes were mapped to genes using SNP2GENE in FUMA[34]. Minor allele frequencies and LD structures were calculated in FUMA using the 1000 Genome phase 3 EUR population[58]. First, the input SNPs were filtered for independent SNPs with $r^2 < 0.6$. For each independent SNP, all known SNPs with $r^2 > 0.6$ with one of the independent significant SNPs were included for further analyses (candidate SNPs). Based on the identified independent significant SNPs, independent lead SNPs were defined if they were independent of each other at $r^2 < 0.1$. In addition, if LD blocks of independent significant SNPs were closely located to each other (< 250 kb), they were merged into one genomic locus. Each genomic locus could thus contain multiple independent significant SNPs and lead SNPs.

Using a positional mapping strategy (10 kb symmetric window), candidate SNPs were mapped to genes in SNP2GENE using MAGMA[33]. These genes were next used in GENE2FUNC with a background of 16573 brain-expressed genes[36] to test for gene set enrichment of the various GO categories (10532 gene sets in MsigDB V7.0 for molecular function, cellular components, biological function). Bonferroni-corrected $p$-value threshold for this step was set to $0.05/10532 = 4.7e-6$.

## Functional relevance of *RBFOX1* associated regions

Using Fisher's exact tests, we calculated odds ratios to quantify the overlap between the sex-differentiated regions in Fig. 4d and each of 17 functional networks defined by Yeo, Krienen et al. [35]. The statistical significance was determined by comparing the observed odds ratio against its null distribution that was built via 10 k spatial permutations/rotations of the cortical surface in Fig. 4d to minimize the potential false positive due to spatial autocorrelation[59,60].

## Comparison with other studies

We compared our sex-stratified $h^2_{SNP}$ results with those of the recent work by Jiang et al.[39]. Using data provided in their Supplementary Table S3, we calculated between-sex Pearson's correlation between sexes for GMV, SA, and CT $h^2_{SNP}$ values as well as performed paired $t$ tests between the sexes.

## Statistics & reproducibility

All statistical analyses using PLINK, GCTA, and FreeSurfer outputs were performed using the R software[56]. Genetic and MRI data were excluded, as described in previous sections. The sample size for this work was chosen by including all individuals with brain MRI data in the latest release of UK Biobank who satisfied the criteria mentioned in "The UK Biobank sample", "Brain MRI phenotypes", and "Quality control" subsections. Randomization and blinding were not used for this work.

## Reporting summary

Further information on research design is available in the Nature Portfolio Reporting Summary linked to this article.

## Data availability

The sex-stratified genome-wide association data generated in this study have been deposited in the https://www.ebi.ac.uk/gwas/ database under accession codes, which may be found in Supplementary Data 22. The imputed genotype data and brain phenotypes in UK Biobank can be obtained by submitting an application at https://www.ukbiobank.ac.uk/enable-your-research/apply-for-access. The SNP-heritability and genetic correlation data generated in this study are provided in the Supplementary Data and Source Data files. Source data are provided in this paper.

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

## Acknowledgements

This research was supported (in part) by the Intramural Research Program of the NIMH (ZIA MH002949-07, ZIC MH002960). TTM was supported by funds from NIH T32HG010464. This work utilized the computational resources of the NIH HPC Biowulf cluster. (http://hpc.nih.gov).

## Author contributions

R.S. and A.R. designed the project. R.S. performed statistical and genetic analyses with feedback from A.R., S.L., T.T.M., and D.M. The overlap between the sex-differentiated regions shown in Fig. 4d and the functional networks were calculated by S.L. UKB Neuroimaging data were processed with FreeSurfer by D.M. The UKB genetic data were hosted and maintained by D.M. and A.T. The manuscript was written by R.S. and A.R. with valuable input from all co-authors.

## Funding

## Competing interests

The authors declare no competing interests.
