## [Peer Review File · Nature Communications]

REVIEWER COMMENTS

Reviewer #1 (Remarks to the Author):

This is an excellent manuscript. It nicely conveys the message that most genetic influences on the many brain phenotypes examined here are shared between males and females, but there are a few differences. The analysis was thoughtfully constructed and the results were clearly reported. The authors also succinctly and accurately described relevant prior literature, and further explained how an understanding of the prior literature informs their analysis strategy (e.g. “We distinguish between these different morphometric properties of the brain because they are known to show distinct genetic architectures”). Kudos! Minor comments are given below, with one that is potentially important (denoted by an asterisk).

Page 2

- Middle of paragraph, line starting with “estimates” - add comma after “genetic correlations” (i.e. Oxford comma improves clarity).
- Probably worth specifying that these are h^2_{SNP} estimates, since h^2 has historically been thought of as twin heritability, and the estimates are usually quite different (i.e. $h^2 > h^2_{\text{SNP}}$)

Page 4

- Beginning of second paragraph, starting with “Here” – change “the UK Biobank sample” to “a sample from the UK Biobank”

Page 5

- Again, probably best to be explicit with h^2_{SNP} throughout, not just h^2 (“SNP-heritability..”).
- How did you do h^2_{SNP} for the x chromosome? Mention the Mallard paper here?
- Middle of second paragraph, line starting with “and insula” – delete parenthesis after “insula”.
- Middle of second paragraph, line starting with “neurodevelopmental disorders” – may want to spell out ASD to say “Autism Spectrum Disorder (ASD)” since it’s the first time the acronym is used
- I agree, but consider moderating this statement: “several of which are likely to reach statistical significance”

Page 6

- Consider adding specific sample size and SNP number in figure 1. Maybe max for both?

- Spell out some abbreviations (MRI, SNP). Personally I don't think you need to spell out PLINK and GCTA.

-

Page 7

- Middle of big paragraph, from "Comparison of male and female h^2 for each phenotype" to "CT ($r = -0.14$, $p = 0.007$,"... - Consider making this more clear: 'no significant regional difference in h^2 for each phenotype' but 'observed some regional variation in magnitude of phenotypic sex differences.' What is the difference between these?

Page 8

- Fig2 D,E,F It might be interesting to give the slopes and tell us whether they are significantly different from 1? Why are the slopes <1 if females tend to have higher h^2 SNP values?

Page 9

Consider adding one more sentence after the following to summarize clearly: "Thus the higher mean VP in males for regional GMV and SA measures in the current analyses as well as reported previously (Ritchie et al., 2018) is accompanied by a higher mean VG, but for a given V_p these traits show a higher VG in females than males which leads to a higher mean h^2 estimate across traits (Figure S4, Supplementary Table S8)."

E.g. "Though males have higher phenotypic variability (variance) on average, ..."

Page 10

- Third line in figure caption, line starting with "and CT (C) – at "low heritability ($p > 0.05$ in for h^2 in either sex)" delete first "in" so it reads: "low heritability ($p > 0.05$ for h^2 in either sex)"

- Fig 3. ABC, poor choice of color scheme. What about the blue-white gradient as in DEF? DEF is very clear. Discriminating among the ABC colors is not intuitive to me.

-

Page 11-12

- Why not interaction tests (i.e. direct test of differences in effect sizes)? I'm not saying you need to do this. Just curious.

Page 12 – I think Fig 4 looks great, very clear and informative

- Standardize whether its "protein coding genes" or "protein-coding genes" – instances of both throughout – first instance I saw is at line 6 of figure 4 caption

Page 14

- "each sex group" -> "both sexes"

- Rephrase “mean cortical thickness is less heritable than total surface area or total GMV” -> “heritability estimates for mean cortical thickness were lower than heritability estimates for total surface area or total GMV”. I mention this in case lower reliability of measurement of cortical thickness is the explanation. If you have ruled this out, disregard this comment and consider mentioning this.
- Add comma: “SNPs which mapped to 505 genes) or whether” -> “SNPs which mapped to 505 genes), or whether”.
- Second paragraph starting with “First” – find -> found
- Second paragraph starting with “Thus” – delete “thus” and just start with “Within both males...” and take out the colon in “see that: mean cortical”
- Middle of second paragraph, line starting with “al., 2021; supp table S5)” – Add comma after “We also find that in each sex”

Page 15

- Ending this paragraph, “cerebellum, putamen and caudate nucleus and lowest for amygdala (Supplementary Table S6) consistent with findings of Satizabal et al. (2019) and Smith et al. (2021).” Can you make any informative comments about how reliable measurement is across these regions and whether or not that likely influenced your findings?
- Middle of second paragraph, line starting with “blood pressure and waist” – add the ending parenthesis after “Gilks et al., 2014)” so it reads: (e.g. for systolic, dystolic... in (Ge...Gilks et al., 2014)).”

Reviewer #2 (Remarks to the Author):

Thank you!

Ciera Stafford

Reviewer #3 (Remarks to the Author):

The authors present important work aiming to identify potential sex differences in common genetic variants shaping brain anatomy, across subcortical and cortical structures using gray matter volume, surface area and cortical thickness. The authors take various approaches to study sex differences in brain anatomy in ~14,000 males and ~16,000 females from the UK Biobank: SNP-based heritability sex differences, between-sex genetic correlations, and sex-stratified GWAS. This is a well-written and well-organized paper, especially given the substantial amount of data to sift through and I think it is a valuable contribution to the field. I enjoyed reading the manuscript. I have some concerns and comments outlined below that I would encourage the authors to integrate/think about to further strengthen the manuscript:

Methods

1) Could the authors comment on their choice of using the Glasser parcellation for the purposes of genetic variant discovery? Given its fine parcellation schema, where many neighbouring regions may be shaped by similar genetic variants, it has been shown that this parcellation has lower discoverability and lower heritability than other cortical atlases (van der Meer et al Cerebral Cortex 2020; <https://doi.org/10.1093/cercor/bhaa146>; Makowski et al., Science 2022).

2) I imagine that many of the 360 regions are highly correlated with each other and do not need to be treated as 360 independent variables in multiple correction testing. I would suggest the authors look at the effective number of independent variables, which takes into consideration the substantial degree of variation that I expect the authors will find in the phenotypic correlation matrix of the HCP parcellation (e.g., matSpD, <https://drive.google.com/open?id=1-r-HWskOD8NfbOG4C4SF1wj8yYze2Zu>). This would apply for the GWAS “strict” threshold (Fig 4A), as well as genetic correlation differences between the sexes as displayed in figure 3D-F. This same approach could be taken for the subcortical structures as well.

3) There are many values not investigated (due to low heritability) in prefrontal cortical thickness regions in particular as seen in Figure 3F, which is a bit of a shame. I wonder again if another atlas would help mitigate this.

4) In general, there are really no visualizations of sex effects on subcortical structures which would be helpful for readers (there is an aseg subcortical atlas implemented within ggseg). I think it is important that more of these subcortical results are integrated into the main manuscript figures, especially given that subcortical structures have the highest heritability in both sexes (cerebellum, putamen, caudate).

5) Could the authors provide rationale for correcting the phenotypes for age²?

6) I appreciate it is hard to do a replication given that sex-stratified analyses would likely be very underpowered in any other sample, but I am curious to know whether some of the top results, especially the rs113078989/RBFOX1 result, replicate in another sample - this could be done in more UKB participants as I am assuming the authors did not use the full UKB sample available to start with based on the numbers they are reporting in their main sample (in methods, the authors note the last download of the data was 2020) .

Discussion

7) It is really interesting and surprising that regional heritability sex differences are largely independent of the magnitude of phenotypic sex differences...do the authors have any additional thoughts or interpretation about this that could be included in their discussion? Do the authors think this is simply a power issue or that the male brain is simply a 'scaled' version of the female brain, but the same genetic variants are at play in both sexes? Or that developmental factors/hormones/puberty are largely driving the phenotypic differences and these relationships may not be captured in the older adult sample of UKB?

8) Could the very high between-sex genetic correlations (as shown by the median values of 1 or close to 1 across GMV, SA, and CT) be one of the main drivers for the lack of genetic variants distinguishing the sexes? Perhaps the authors could comment on this in the discussion.

9) Is there anything meaningful that could be said about the pattern of regions in Fig 4D that show sex differences in the top uncovered SNP? Nothing jumps out at me immediately but perhaps there are some circuits/anatomical connections here to comment upon.

Plots

10) Would it be possible for the authors to use some colours for the scatterplots, for instance different colours for lobes of the brain just to get a sense of whether there is some structural grouping to the scatterplots, and to help orient the readers a bit more to regions of the brain in the scatterplots.

11) Figure 2 (D, E, F)- it is unclear what the black solid line represents

Reviewer #4 (Remarks to the Author):

Shafee et al. have examined potential sex differences in the genetic basis of human brain anatomy, utilizing the expansive dataset of the UK Biobank. The main findings reinforce the notion that common variants that influence human brain anatomy are largely consistent across sexes, while highlighting specific areas where sex differences are pronounced. The study's strength lies in its rigorous approach to assessing over 1,100 neuroanatomical traits, providing a nuanced understanding of how genetics may influence brain structure in a sex-specific way.

One of the key findings is that despite the general concordance between males and females, the higher mean heritability observed in females for cortical gray matter volumes and surface areas. Secondly, the parietal cortex and insula show substantial variation in genetic correlations across sexes. Additional intriguing finding is the discovery of a male-specific effect of a common variant mapping to the RBFOX1 gene, linked with its association with multiple male-biased neuropsychiatric disorders. This finding opens new research avenues for understanding the genetic basis of these disorders and their sex-specific manifestations.

Major comments:

1. Figure 3C. Based on the color scheme, many CT regions show r_g less than 0.5 and some closer to 0. But, likelihood ratio test p-values listed in Figure 3F only highlights posterior insular. What would be the reason?
2. Figure 3I. It looks like higher sex-differences in surface areas are correlated with more significant genetic correlations between sexes, which seems to be counter-intuitive. How would you interpret the findings?
3. Figure 4E. Please indicate whether two groups show significant mean differences in the graph with a p-value.
4. Figure 4F. Please provide annotations that explain what the numeric values represent.
5. Implication of RBFOX1 is intriguing. Is there any functional impact of the implicated SNP or the region related to the gene (e.g., enhancer)?
6. Similarly, I would suggest evaluating potential differences of RBFOX1 gene expression between males and females (e.g., PrediXcan). The gene is a master regulator, therefore evaluation of downstream genes may be informative as well.
7. Few discussions are provided for the second GWAS-sig SNP on the chromosome X. Considering its location on the sex chromosome and potential dosage-dependent impact, as the authors described as one of main sex-biased effects, more discussion of the region is recommended.
8. Compared to the authors' previous work (Liu 2020), few results are presented related to sex chromosomes. Authors need to compare and/or evaluate their X-chromosome findings related to Jiang

(2023)'s UKBB neuroimaging genomics study.

<https://www.medrxiv.org/content/10.1101/2023.08.30.23294848v1>

9. Further discussions are also needed about how to two brain regions, superior parietal Brodmann area 5 and posterior insula, may exhibit sex-specific manifestations and their potential evolutionary benefits.

Minor comments:

1. This is very confusing statement. Please revise it clearly.

“Thus the higher mean V P in males for regional GMV and SA measures in the current analyses as well as reported previously (Ritchie et al., 2018) is accompanied by a higher mean V G , but for a given V p these traits show a higher V G in females than males which leads to a higher mean h 2 estimate across traits (Figure S4, Supplementary Table S8).”

Reviewer #1 (Remarks to the Author):

Comment 1

This is an excellent manuscript. It nicely conveys the message that most genetic influences on the many brain phenotypes examined here are shared between males and females, but there are a few differences. The analysis was thoughtfully constructed and the results were clearly reported. The authors also succinctly and accurately described relevant prior literature, and further explained how an understanding of the prior literature informs their analysis strategy (e.g. “We distinguish between these different morphometric properties of the brain because they are known to show distinct genetic architectures”). Kudos! Minor comments are given below, with one that is potentially important (denoted by an asterisk).

Response 1

We thank the reviewer for their encouraging comments about our work and helpful suggestions for improvement - which we have addressed as detailed in the comments below.

Comment 2

Middle of paragraph, line starting with “estimates” - add comma after “genetic correlations” (i.e. Oxford comma improves clarity).

Response 2

We have changed this line to include the comma as suggested (line 26).

Comment 3

Probably worth specifying that these are h²SNP estimates, since h² has historically been thought of as twin heritability, and the estimates are usually quite different (i.e. h²>h²SNP)

Response 3

This has now been specified as requested by adding the clarifying line below.

Line 88: *"It should be noted that estimated SNP-heritability is usually less than heritability estimates from twin studies."*

Comment 4

Page 4, Beginning of second paragraph, starting with “Here” – change “the UK Biobank sample” to “a sample from the UK Biobank”

Response 4

We have made this change (line 77).

Comment 5

Page 5. Again, probably best to be explicit with h²SNP throughout, not just h² (“SNP-heritability.. “).

Response 5

We have now changed all h^2 in the text as well as in the figures to h^2_{SNP} .

Comment 6

How did you do h^2_{SNP} for the x chromosome? Mention the Mallard paper here?

Response 6

The Mallard et al paper is mentioned when addressing X-linked h^2 estimation in our Methods and Results sections:

lines 470-472: "For each phenotype X-linked h^2 was estimated using the best-fit dosage compensation model from (Mallard et al., 2021)".

lines 129-131: "X-linked heritability was estimated using the best-fit dosage compensation models (full dosage compensation/no dosage compensation/equal variance) from Mallard et al. ²² for each phenotype."

Comment 7

Middle of second paragraph, line starting with "and insula" – delete parenthesis after "insula".

Response 7

We have deleted the parenthesis (line 100).

Comment 8

Middle of second paragraph, line starting with "neurodevelopmental disorders" – may want to spell out ASD to say "Autism Spectrum Disorder (ASD)" since it's the first time the acronym is used

Response 8:

We have now spelled out ASD at first mention as requested (line 103).

Comment 9

I agree, but consider moderating this statement: "several of which are likely to reach statistical significance"

Response 9

We have now modified the sentence to read:

line 105: "*and some of these nominal associations may reach statistical significance with expanding sample sizes (Visscher et al., 2017).*"

Comment 10

Page 6. Consider adding specific sample size and SNP number in figure 1. Maybe max for both?

Response 10

We have now added the exact numbers in Figure 1.

Comment 11

Spell out some abbreviations (MRI, SNP). Personally I don't think you need to spell out PLINK and GCTA.

Response 11

We have now spelled out MRI (line 54) and SNP (line 87).

Comment 12

Page 7, Middle of big paragraph, from "Comparison of male and female h^2 for each phenotype" to "CT ($r = -0.14$, $p = 0.007$,"... - Consider making this more clear: 'no significant regional difference in h^2 for each phenotype' but 'observed some regional variation in magnitude of phenotypic sex differences.' What is the difference between these?

Response 12

We have now clarified these statements as requested and the new text is reproduced below.

Lines 134-136: "*No statistically significant between-sex difference in h^2 was observed for any of these regional h^2 estimates (Methods) after correcting for multiple-testing (all $p > 1.4e-4$, Supplementary Tables S2, S3, S4). Additionally, the variation in the magnitude of sex-differences in h^2 (Figure 2g,h,i) was unrelated to the magnitude of phenotypic sex differences (calculated as described in Methods) for GMV and SA, but showed a weakly negative correlation with regional variation in phenotypic sex-differences for CT ($r = -0.14$, $p = 0.007$, Supplementary Figure S2).*"

Comment 13:

Page 8, Fig2 D,E,F It might be interesting to give the slopes and tell us whether they are significantly different from 1? Why are the slopes < 1 if females tend to have higher h^2 SNP values?

Response 13

To test for sex differences in mean h^2 , we statistically compared the distribution of h^2 values between males and females using both t-tests and the Wilcoxon rank test. This result is most closely aligned to the orthogonal deviation of the fit lines in 2d-2f from the diagonal identity line. These deviations are positive (i.e. $F > M$) for GMV and SA around the modal values of h^2 estimates - echoing the results of the aforementioned t- and Wilcoxon rank tests. The slopes of the fit lines in 2d-2f and their deviation from 1 are related to a different concept which our manuscript does not address: whether any difference in h^2 estimates between males and females changes monotonically across the range of observed h^2 estimates. Given this, we have elected to not add in estimates

of the slope - although we agree that this sort of analysis could have interesting broader applications in other analytic contexts. To prevent confusion, we have now removed the fit lines from Fig 2d-f.

Comment 14

Page 9, Consider adding one more sentence after the following to summarize clearly: “Thus the higher mean VP in males for regional GMV and SA measures in the current analyses as well as reported previously (Ritchie et al., 2018) is accompanied by a higher mean VG, but for a given Vp these traits show a higher VG in females than males which leads to a higher mean h2 estimate across traits (Figure S4, Supplementary Table S8).”

E.g. “Though males have higher phenotypic variability (variance) on average, ...”

Response 14

We have now modified this passage as suggested to read:

Line 167-171: “Thus, consistent with previously reported results (Ritchie et al., 2018) males showed higher phenotypic and genetic variance in GMV and SA averaged across all brain regions even though on average, heritability, which is the ratio of the two, was higher in females (Supplementary Fig. S4, Supplementary Table S8).”

Comment 15

Page 10, Third line in figure caption, line starting with “and CT (C) – at “low heritability ($p > 0.05$ in for h2 in either sex)” delete first “in” so it reads: “low heritability ($p > 0.05$ for h2 in either sex)”

Response 15

We have now made this change.

Comment 16

Fig 3. ABC, poor choice of color scheme. What about the blue-white gradient as in DEF? DEF is very clear. Discriminating among the ABC colors is not intuitive to me.

Response 16

We have now changed the color scheme to match panels D,E,F as suggested.

Comment 17:

Page 11-12, Why not interaction tests (i.e. direct test of differences in effect sizes)? I’m not saying you need to do this. Just curious.

Response 17

We chose this approach in order to: (i) take into account possible sex-dependent covariate effects which would not be accounted for in a SNP x sex interaction test in PLINK and, (ii) generate sex-specific manhattan plots for each phenotype for qualitative assessment of the association tests in each sex.

Comment 18

Page 12 – I think Fig 4 looks great, very clear and informative. Standardize whether its “protein coding genes” or “protein-coding genes” – instances of both throughout – first instance I saw is at line 6 of figure 4 caption

Response 18

We have now used "protein-coding gene" throughout the manuscript.

Comment 19

Page 14, “each sex group” -> “both sexes”

Response 19:

We have now changed this text to read “both sexes” as suggested (line 287).

Comment 20

Rephrase “mean cortical thickness is less heritable than total surface area or total GMV” -> “heritability estimates for mean cortical thickness were lower than heritability estimates for total surface area or total GMV”. I mention this in case lower reliability of measurement of cortical thickness is the explanation. If you have ruled this out, disregard this comment and consider mentioning this.

Response 20

We agree with the reviewer that one potential contributor to the lower estimated h^2 for CT might be lower test-retest reliability of CT compared to SA or subcortical volumes as shown in Haddad et al. 2022. We have now changed the sentence as suggested by the reviewer.

lines 287-289: *"heritability estimates for mean cortical thickness were lower than heritability estimates for total surface area or total GMV"*

Comment 21

Add comma: “SNPs which mapped to 505 genes) or whether” -> “SNPs which mapped to 505 genes), or whether”.

Response 21

This comma has now been added (line 277).

Comment 22

Second paragraph starting with “First” – find -> found

Response 22

We have now made the change (and also other ones consistent with it) in the discussion section.

Comment 23

Second paragraph starting with “Thus” – delete “thus” and just start with “Within both males...” and take out the colon in “see that: mean cortical”

Response 23

We have now made the change suggested by the reviewer (line 277).

Comment 24

Middle of second paragraph, line starting with “al., 2021; supp table S5)” – Add comma after “We also find that in each sex”

Response 24

We have now added the comma (line 289).

Comment 25

Page 15, Ending this paragraph, “cerebellum, putamen and caudate nucleus and lowest for amygdala (Supplementary Table S6) consistent with findings of Satizabal et al. (2019) and Smith et al. (2021).” Can you make any informative comments about how reliable measurement is across these regions and whether or not that likely influenced your findings?

Response 25

We thank the reviewer for raising this interesting interpretation, and we agree that regional differences in reliability of anatomical measurement would indeed have the potential to shape regional differences in heritability estimates. Motivated by the reviewer’s comment we looked up published reliability results for UKB subcortical volumes in Haddad et al. (2022, Human Brain Mapping), but these did not track cleanly with our observed regional differences in subcortical heritability. For example, the pallidum has relatively low reliability in Haddad, but relatively high heritability in our analyses. Although the focus of our work was sex differences in heritability, we agree that future studies on the relationship between measurement properties and heritability are important for the field.

Comment 26

Middle of second paragraph, line starting with “blood pressure and waist” – add the ending parenthesis after “Gilks et al., 2014)” so it reads: (e.g. for systolic, dystolic... in (Ge...Gilks et al., 2014)).”

Response 26

We have now edited that sentence as suggested by the reviewer (line 313).

Reviewer #2 (Remarks to the Author):

Thank you!
Ciera Stafford

Response;
Thank you for your help with the revision process.

Reviewer #3 (Remarks to the Author):

Comment 1

The authors present important work aiming to identify potential sex differences in common genetic variants shaping brain anatomy, across subcortical and cortical structures using gray matter volume, surface area and cortical thickness. The authors take various approaches to study sex differences in brain anatomy in ~14,000 males and ~16,000 females from the UK Biobank: SNP-based heritability sex differences, between-sex genetic correlations, and sex-stratified GWAS. This is a well-written and well-organized paper, especially given the substantial amount of data to sift through and I think it is a valuable contribution to the field. I enjoyed reading the manuscript. I have some concerns and comments outlined below that I would encourage the authors to integrate/think about to further strengthen the manuscript:

Response 1

We thank the reviewer for the careful and helpful review of our work. We have modified our manuscript to address all the specific issues raised as detailed below.

Comment 2:

Methods. Could the authors comment on their choice of using the Glasser parcellation for the purposes of genetic variant discovery? Given its fine parcellation schema, where many neighbouring regions may be shaped by similar genetic variants, it has been shown that this parcellation has lower discoverability and lower heritability than other cortical atlases (van der Meer et al Cerebral Cortex 2020; <https://doi.org/10.1093/cercor/bhaa146>; Makowski et al., Science 2022).

Response 2

We thank the reviewer for raising this important question regarding our choice of parcellation scheme. We used the Glasser et al parcellation to bring our work in line with other recent imaging-genetic publications by our own group (Mallard et al, Nature Neuroscience, 2021) and others (Warrier et al, Nature Genetics, 2023) - with the aim of facilitating integration of results across imaging genetic studies and comparison of results from imaging-genetics to the growing number of non-genetic neuroimaging studies that are rooted in the Glasser parcellation (e.g. Sydnor et al, Neuron, 2021). We were also motivated by the idea that the multimodal imaging data underpinning the Glasser parcellation helps to maximize the capacity of neuroimaging data to inform methodology of imaging-genetics - although we acknowledge that this is an empirical question. We have therefore implemented Reviewer's recommendation for additional analyses in Comment 3 below. Our findings (see Response 3) suggest that accounting for the interdependencies between Glasser parcels does not boost discovery of sex-biased genetic effects.

Comment 3

2) I imagine that many of the 360 regions are highly correlated with each other and do not need to be treated as 360 independent variables in multiple correction testing. I would suggest the authors look at the effective number of independent variables, which takes into consideration the substantial degree of variation that I expect the authors will find in the phenotypic correlation matrix of the HCP parcellation (e.g., matSpD, <https://drive.google.com/open?id=1-r-HWsKOD8NfbOG4C4SF1wjj8yYze2Zu>). This would apply for the GWAS “strict” threshold (Fig 4A), as well as genetic correlation differences between the sexes as displayed in figure 3D-F. This same approach could be taken for the subcortical structures as well.

Response 3

We thank the reviewer for this excellent suggestion which we have implemented. Specifically, we have now calculated less stringent p-value thresholds using the reviewer's suggestion (matSpD.R). We found the number of effective tests to be 223 for SA, 299 for CT and 253 for GMV. Using multiple-testing corrections corresponding to these numbers did not increase the yield of detected: (i) sex difference in h^2 , (ii) SNP with significant sex-biased effect (so only two SNPs shown in Figure 4 as before), or (iii) deviations of between-sex r_g from 1. Given this outcome, we have elected to not include these additional results in our paper, but we would be happy to do so if need be.

Comment 4

There are many values not investigated (due to low heritability) in prefrontal cortical thickness regions in particular as seen in Figure 3F, which is a bit of a shame. I wonder again if another atlas would help mitigate this.

Response 4

We have now added a new Figure in the Supplement (supplementary Fig. S5) which is reproduced below and shows the point estimates of r_g for all regions. For these additional regions the error bars or estimated r_g are large and the LRT results for $r_g < 1$ remain unchanged. Please see Responses 2 and 3 above regarding the issue of parcellation selection.

Comment 5

In general, there are really no visualizations of sex effects on subcortical structures which would be helpful for readers (there is an aseg subcortical atlas implemented within ggseg). I think it is important that more of these subcortical results are integrated into the main manuscript figures, especially given that subcortical structures have the highest heritability in both sexes (cerebellum, putamen, caudate).

Response 5

We have now added a panel in Supplementary Fig. S3 (reproduced below) showing the subcortical structure h2 in males and females separately. Since there was no significant sex difference in h2 and also no significant deviation from $r_g = 1$ for these structures, we did not include the results in the figures in the main text.

Comment 6

Could the authors provide rationale for correcting the phenotypes for age²?

Response 6

Our usage of age² is guided by previous studies which used UKB MRI data, for example, Elliott et al. 2018 (Nature) and Mallard et al. 2021 (Nat Neuroscience). The significant variance captured by age² has also been shown in Alfaro-Almargo et al. 2021 (Neuroimage).

Comment 7

I appreciate it is hard to do a replication given that sex-stratified analyses would likely be very underpowered in any other sample, but I am curious to know whether some of the top results, especially the rs113078989/RBFOX1 result, replicate in another sample - this could be done in more UKB participants as I am assuming the authors did not use the full UKB sample available to start with based on the numbers they are reporting in their main sample (in methods, the authors note the last download of the data was 2020).

Response 7

Our analysis uses all UKB participants of European ancestry with neuroimaging and genetic data that passed QA/QC. As such, we do not have any other data that would enable an interpretable attempt at replication unfortunately. We agree however that replication will be an important goal for future work when suitable data are available.

Comment 8

It is really interesting and surprising that regional heritability sex differences are largely independent of the magnitude of phenotypic sex differences...do the authors have any additional thoughts or interpretation about this that could be included in their discussion? Do the authors think this is simply a power issue or that the male brain is simply a 'scaled' version of the female brain, but the same genetic variants are at play in both sexes? Or that developmental factors/hormones/puberty are largely driving the phenotypic differences and these relationships may not be captured in the older adult sample of UKB?

Response 8

We thank the reviewer for highlighting this interest point for expanded consideration in Discussion. We have offered some additional thoughts within the Discussion text as suggested and this new passage is reproduced below:

Line 335-340: "Thus, in general, sex differences in the mean value of anatomical traits appear to be achieved by mechanisms that do not modify the collective magnitude, or genome wide distribution, of common variant influences on trait variation within each sex. Such mechanisms could potentially reflect sex difference in gene dosage or environmental exposures - which may both exert their effects in a developmentally dynamic manner."

Comment 9

Could the very high between-sex genetic correlations (as shown by the median values of 1 or close to 1 across GMV, SA, and CT) be one of the main drivers for the lack of genetic variants distinguishing the sexes? Perhaps the authors could comment on this in the discussion.

Response 9

These two results are consistent with each other in that they both capture the same phenomenon of overall similarity in the common genetic architecture of brain anatomy in males and females.

Comment 10

Is there anything meaningful that could be said about the pattern of regions in Fig 4D that show sex differences in the top uncovered SNP? Nothing jumps out at me immediately but perhaps there are some circuits/anatomical connections here to comment upon.

Response 10

We have sought to address this interesting question by quantifying overlap between the regions in 4D with the Yeo - Krienen 7 and 17 network parcellations (Yeo, Krienen, et al. 2011). There is a trend for M > F regions to lie within a component of the default mode network (DMN-2) with spin-test p-value of 0.07. The F > M regions do not show any

suggestive overlap with a functional network. We have now added these results as a supplementary table (Supplementary Table S21) and have referred to the associated Methods and Results in the text as reproduced below.

Line 263-266: "A test for potential enrichment functional relevance of these regions using spin tests with the the 17 networks of Yeo, Krienen et al. ³⁵ pointed to suggestive overlap with the default mode network (DMN-2) for the M>F regions and no overlap with the functional networks for the F>M regions (Table S21)."

Line 533-537: "Using Fisher's exact tests, we calculated odds ratios to quantify the overlap between male- and female-biased regions in Fig 4d and each of 17 functional networks defined by Yeo, Krienen et al. ³⁵. The statistical significance was determined by comparing the observed odds ratio against its null distribution that was built via 10k spatial permutations/rotations of the cortical surface in Fig 4d to minimize the potential false positive due to spatial autocorrelation ^{58,59}."

Comment 11

Plots - Would it be possible for the authors to use some colours for the scatterplots, for instance different colours for lobes of the brain just to get a sense of whether there is some structural grouping to the scatterplots, and to help orient the readers a bit more to regions of the brain in the scatterplots.

Response 11

We have now applied a lobar assignment of Glasser parcels to provide the requested information, and the modified scatterplot panels are reproduced below.

Figure 2 (d,e,f)

Figure 3 (g,h,i)

Supplementary Figure S2 (d,e,f)

Comment 12

Figure 2 (D, E, F)- it is unclear what the black solid line represents

Response12

We have now clarified this in the figure caption. "The solid black line shows a line with a slope of 1."

Reviewer #4 (Remarks to the Author):

Comment 1:

Shafee et al. have examined potential sex differences in the genetic basis of human brain anatomy, utilizing the expansive dataset of the UK Biobank. The main findings reinforce the notion that common variants that influence human brain anatomy are largely consistent across sexes, while highlighting specific areas where sex differences are pronounced. The study's strength lies in its rigorous approach to assessing over 1,100 neuroanatomical traits, providing a nuanced understanding of how genetics may influence brain structure in a sex-specific way.

One of the key findings is that despite the general concordance between males and females, the higher mean heritability observed in females for cortical gray matter volumes and surface areas. Secondly, the parietal cortex and insula show substantial variation in genetic correlations across sexes. Additional intriguing finding is the discovery of a male-specific effect of a common variant mapping to the RBF0X1 gene, linked with its association with multiple male-biased neuropsychiatric disorders. This finding opens new research avenues for understanding the genetic basis of these disorders and their sex-specific manifestations.

Response 1:

We thank the reviewer for the kind and thoughtful summary as well as their helpful comments for improvement.

Comment 2:

Figure 3C. Based on the color scheme, many CT regions show r_g less than 0.5 and some closer to 0. But, likelihood ratio test p-values listed in Figure 3F only highlights posterior insular. What would be the reason?

Response 2

The LRT reflects the differences in the log-likelihood values of the two models ($r_g = 0.5$, for example and $r_g = 1$) and the p-value is deduced from the LRT. Even when a point estimate of r_g is low the associated error may be large which in turn can affect the eventual LRT p-value. For example, in one of the regions (CT), $r_g = -0.01$ (0.22) and LRT statistics is 8.36 and p-value is $1.9e-3$. So even though the point estimate is effectively 0, this region still did not make it to the final list of regions with $r_g < 1$ with Bonferroni correction.

Comment 3

Figure 3I. It looks like higher sex-differences in surface areas are correlated with more significant genetic correlations between sexes, which seems to be counter-intuitive. How would you interpret the findings?

Response 3

While the correlation between the two quantities is positive, this actually reflects the fact that those regions that have the lowest sex difference (Male-Female = 0) are the ones with r_g close to 1 (indicated by axis values). The regions with highest sex differences (most negative male-female SA) show the lowest genetic correlation.

Comment 4

Figure 4E. Please indicate whether two groups show significant mean differences in the graph with a p-value.

Response 4

We thank the reviewer for pointing this out. The two groups show statistically significant GMV differences with $p = 8e-6$ (higher in CT compared to TT). This has now been added to Figure 4.

Comment 5

Figure 4F. Please provide annotations that explain what the numeric values represent.

Response 5

The unit has been added to Figure 4f and also mentioned in the caption.

Comment 6

Implication of RBFOX1 is intriguing. Is there any functional impact of the implicated SNP or the region related to the gene (e.g., enhancer)?

Response 6

This is an excellent point. The SNP rs113078989 is intronic to the gene *RBFOX1*, which we now mention in the text. The SNP was not found to be in an enhancer/promoter site of *RBFOX1* (www.genome.ucsc.edu and www.encodeproject.org). We have now added this information to the main text and in Supplementary Fig. S6a (shown below).

line 251-253: *"The autosomal SNP chr16:rs113078989 mapped to the intronic region of the gene RBFOX1 whereas chrX:rs747862348 was not within 100 kb of any gene and was not within any enhancer or promoter region (www.ucsc.edu)."*

Comment 7

Similarly, I would suggest evaluating potential differences of RBFOX1 gene expression between males and females (e.g., PrediXcan). The gene is a master regulator, therefore evaluation of downstream genes may be informative as well.

Response 7

Following the reviewer's suggestion our literature search showed that *RBFOX1* has been reported to be female-biased in expression in brain tissue by Naqvi et al, (Science, 2019 ; Supplementary Table 4 in their paper). We have also now checked and determined that the SNP implicated by our sex-stratified GWAS has not been reported as a sex-biased eQTL by Olivia et al, Science, 2020 - although we note that power is currently low for such analyses even in GTEx. We now added this information to our manuscript. However, we elected to focus this new text on RBFOX1 (text additions reproduced below) because the optimal approach for evaluating gene regulated by FOXP1 in the context of our analyses is not clear, and presents many interesting challenges such as the identification of regulated genes in the regions were we detect a sex-biased effect of the SNP, determining in regulation is itself sex-biased, and designing a statistical test or sex-biased GWAS signal at the gene-set level.

Lines 350-352: "It has also been found to show female-biased expression (Naqvi et al. 2019). The RBFOX1 implicating SNP rs113078989 has not been reported as a sex-biased eQTL by Olivia et al, Science, 2020 - but we note that power is currently low for such analyses even in GTEx."

Comment 8

Few discussions are provided for the second GWAS-sig SNP on the chromosome X. Considering its location on the sex chromosome and potential dosage-dependent impact, as the authors described as one of main sex-biased effects, more discussion of the region is recommended.

Response 8

The SNP rs747862348 did not map to any gene when a 10kbp window is used for positional gene mapping. Extending the window to 100 kb did not change the result. Additionally, the SNP does not fall within any enhancer or promoter region (www.genome.ucsc.edu and www.encodeproject.org, also Supplementary Fig. 6b, shown below). We have now added this information in the text:

*Lines 251-253: "chrX:rs747862348 was not within 100 kb of any gene and was not within any enhancer or promoter region (www.ucsc.edu, **Supplementary Fig. S6**)."*

Comment 9

Compared to the authors' previous work (Liu 2020), few results are presented related to sex chromosomes. Authors need to compare and/or evaluate their X-chromosome findings related to Jiang (2023)'s UKBB neuroimaging genomics study.

<https://www.medrxiv.org/content/10.1101/2023.08.30.23294848v1>

Response 9:

We thank the reviewer for highlighting the excellent Jiang et al preprint. Given that this manuscript uses the Desikan parcellation in contrast to the Glasser et al that is used in our current report and Mallard et al, we were not unfortunately able to carry out a formal quantitative comparison of results. The X-chromosome focused results from Jiang et al are most suitable for comparison with the results of Mallard et al (which focuses on the X-chromosome) rather than our current report which does not map enrichment of X-chromosome influences. However, Jiang et al do provide results for a sex-stratified analyses of h^2 , and we have now included comparison with their work in the Discussion section of the main text. The methods section now also has an extra section, "comparison with other studies", corresponding to this comparison.

lines 299-306: *"In their recent work focusing on the effects of the X-chromosome on brain anatomy Jiang et al.³⁹ reported sex-stratified h^2_{SNP} for CT, GMV and SA in the UK Biobank. Due to the difference in choice of parcellation we are not able to directly compare our results with their findings. However, their results also show high correlation between male and female h^2_{SNP} for all three categories: $r = 0.82$, 0.88 and 0.71 for GMV, SA and CT, respectively. Paired t-tests using their sex-stratified h^2_{SNP} values showed male-bias in CT ($p = 0.00017$) and female-bias in SA ($p = 0.00071$) - the latter in agreement with our findings although we did not detect any sex-bias in h^2_{SNP} of CT."*

Comment 10:

Further discussions are also needed about how two brain regions, superior parietal Brodmann area 5 and posterior insula, may exhibit sex-specific manifestations and their potential evolutionary benefits.

Response 10

We highlight how these two brain regions are notably also foci of sex differences in mean anatomical traits (SA is female biased and posterior insula CT is male biased) - and this text is reproduced below. We agree that the question of potential evolutionary implications for sex-biased genetic regulation of these two brain regions is a fascinating one, but there are unfortunately limited data to constrain speculation on this question. We now provide expanded discussion of this issue as requested by the reviewer, and the new Discussion text is reproduced below:

Lines 328-333: *"Also, both of the cortical phenotypes showing between-sex rg values significantly below 1 - BA5 surface area and insula cortical thickness - also showed significant phenotypic sex differences in their means. The lower between-sex rg in these regions suggests that differential genetic regulation of anatomical variation may contribute to sex-differences in the mean trait value attained - although it remains unclear if these differences carry any functional consequences (DeCasien et al, Biol. Sex Differ 2022)."*

Comment 11:

This is very confusing statement. Please revise it clearly.

Thus the higher mean V_P in males for regional GMV and SA measures in the current analyses as well as reported previously (Ritchie et al., 2018) is accompanied by a higher mean V_G , but for a given V_P these traits show a higher V_G in females than males which leads to a higher mean h^2 estimate across traits (Supplementary Figure S4, Supplementary Table S8).

Response 11

We have now edited that section which now reads:

Lines 164-171: *"Paired t-tests (as well as nonparametric Wilcoxon rank tests) between the sexes indicated higher mean V_G , V_P and V_E in males compared to females in GMV (t -stat > 5.9 and $p < 5e-9$ for all three) and SA (t -stat > 7, $p < 5.4e-12$ for for all three), but not CT (all three t -test $p > 0.05$) (Supplementary Table S7). Thus, consistent with previously reported results (Ritchie et al., 2018) males showed higher phenotypic and genetic variance in GMV and SA averaged across all brain regions even though on average, heritability, which is the ratio of the two, was higher in females (Supplementary Fig. S4, Supplementary Table S8)."*

REVIEWERS' COMMENTS

Reviewer #2 (Remarks to the Author):

Thank you for the informative revisions. No further comments.

- Ciera Stafford

Reviewer #3 (Remarks to the Author):

The authors have done an excellent job addressing reviews. I have no further comments.

Reviewer #4 (Remarks to the Author):

The authors addressed all the questions I had.